# Influence of Climate and Land Cover/Use Change on Water Balance: An Approach to Individual and Combined Effects

Rebeca Martínez-Retureta [1,2,*], Mauricio Aguayo [1,*], Norberto J. Abreu [3,4,*], Roberto Urrutia [1,2], Cristian Echeverría [5], Octavio Lagos [2,6], Lien Rodríguez-López [7], Iongel Duran-Llacer [8] and Ricardo O. Barra [1,2]

1　Centro EULA, Facultad de Ciencias Ambientales, Universidad de Concepción, Concepción 4070386, Chile; rurrutia@udec.cl (R.U.); ricbarra@udec.cl (R.O.B.)
2　Centro de Recursos Hídricos para la Agricultura y la Minería—CRHIAM, Centro FONDAP ANID, Victoria 1295, Concepción 4030000, Chile; octaviolagos@udec.cl
3　Departamento de Ingeniería Química, Facultad de Ingeniería y Ciencias, Universidad de la Frontera, Casilla 54-D, Temuco 4780000, Chile
4　Centro de Manejo de Residuos y Bioenergía, Núcleo Científico y Tecnológico en Biorecursos, Universidad de la Frontera, Casilla 54-D, Temuco 4780000, Chile
5　Laboratorio de Ecología de Paisaje, Facultad de Ciencias Forestales, Universidad de Concepción, Concepción 4070386, Chile; cristian.echeverria@udec.cl
6　Departamento de Recursos Hídricos, Facultad de Ingeniería Agrícola, Universidad de Concepción, Chillán 3812120, Chile
7　Facultad de Ingeniería, Arquitectura y Diseño, Universidad San Sebastián, Lientur 1457, Concepción 4030000, Chile; lien.rodriguez@uss.cl
8　Hémera Centro de Observación de la Tierra, Facultad de Ciencias, Ingeniería y Tecnología, Universidad Mayor, Camino La Pirámide 5750, Huechuraba, Santiago 8580745, Chile; iongel.duran@gmail.com
*　Correspondence: rebecmartinez@udec.cl (R.M.-R.); maaguayo@udec.cl (M.A.); norberto.abreu@ufrontera.cl (N.J.A.)

**Abstract:** Land use/cover change (LUCC) and climate change (CC) affect water resource availability as they alter important hydrological processes. Mentioned factors modify the magnitude of surface runoff, groundwater recharge, and river flow among other parameters. In the present work, changes that occurred in the recent decades at the Quino and Muco river watersheds in the south-central zone of Chile were evaluated to predict future cover/use changes considering a forest expansion scenario according to Chilean regulations. In this way an expansion by 42.3 km$^2$ and 52.7 km$^2$ at Quino and Muco watersheds, respectively, was predicted, reaching a watersheds' occupation of 35.4% and 22.3% in 2051. Additionally, the local climatic model RegCM4-MPI-ESM-MR was used considering periods from 2020–2049 and 2050–2079, under the RCP 8.5 scenario. Finally, the SWAT model was applied to assess the hydrological response of both watersheds facing the considered forcing factors. Five scenarios were determined to evaluate the LUCC and CC individual and combined effects. The results depict a higher sensitivity of the watersheds to CC impacts, where an increase of evapotranspiration, with a lessening of percolation, surface flow, lateral flow, and groundwater flow, triggered a water yield (WYLD) decrease in all predicted scenarios. However, when both global changes act synergistically, the WYLD decreases considerably with reductions of 109.8 mm and 123.3 mm at the Quino and Muco watersheds, respectively, in the most extreme simulated scenario. This water scarcity context highlights the necessity to promote land use management strategies to counteract the imminent effects of CC in the watersheds.

**Keywords:** climate change; hydrological cycle; land cover/use change; SWAT model

## 1. Introduction

Along with and caused in part by global and regional economic development, land use/cover change (LUCC) and climate change (CC) have affected water resources, involv-

ing critical adverse future scenarios [1–6]. Rapid human population growth over the last century and the consequent increase in water and food requirements, together with high rainfall variability and extreme hydrological events as part of the CC, have undermined the availability of biophysical resources [7,8].

On the one hand, anthropogenic activities such as agriculture, industry, and transportation, along with various socio-economic, political, and institutional factors, have involved important changes in land use. In Europe, for example, the landscape has changed radically due to the political and socio-economic changes that occurred during the first half of the 19th century [9]. In addition, in some regions of Africa, the expansion of agriculture influenced by the fast population growth has been recognized as the main driver of LUCC [10]. Furthermore, in developed regions such as the United States of America, the European Union, and Japan, as well as in developing countries such as India, Vietnam, China, and the Philippines, the forest transition phenomenon has been clearly evidenced [11]. While some regions of countries such as Pakistan were forced to abandon their farmable land due to water scarcity and declining farm incomes [12]. In this way, LUCC is no longer a local environmental problem but is becoming a global trend involving changes in the terrestrial ecosystem, ecological service structure and function, and biodiversity, among others [13,14].

On the other one, CC imposes additional pressure onthe availability and accessibility of water resources affecting directly water partition within the watersheds [8,15,16]. It could be expected that these global changes may generate adverse environmental effects, which increase the relevance of determining their effect on the hydrological processes at different temporal–spatial scales.

In this sense, elements such as demography, institutions, technology, and macroeconomic activities, among others, cause important alterations in land use, consequently affecting hydrological systems, both at the watershed and regional scales [17,18]. Additionally, LUCC is considered one of the main forcing factors among terrestrial and atmospheric components of the hydrological cycle as it is directly related to water quantity and hydrological processes [19–21]. In addition, CC affects the water cycle by changing the temporal–spatial pattern of precipitation, subsequently modifying watershed runoff and flow behavior [22,23]. Therefore, the rapid LUCC, together with CC, could lead to increased hydrological impacts by altering the magnitude of the hydrological process in watersheds [24,25].

Few studies have explicitly recognized the combined effect of LUCC and CC [22,26–29]. Due to the magnitude of the expected potential impacts, this research area has become very relevant [28,30–33]. The above research reports have shown that the hydrological response of watersheds to LUCC varies with climate and with the physical-geographical characteristics of the area. In addition, Qi et al. [34] suggest that future hydrological changes and LUCC should be site-specific. They also state the importance of taking into account the climate variability to control the hydrological process of the watershed. Nevertheless, most of the studies concerning LUCC have not considered the effects of CC and vice versa. Detailed information abouttheir interaction remains unavailable and relevant information on their impacts at different spatial scales, useful for local actors, farmers, and decisionmakers, remains limited [35]. A broad understanding of the impacts of LUCC and CC is essential to ease their adverse effects onthe water resources through integrated watershed management for healthy ecosystems [35]. Therefore, it is necessary to develop studies considering both the individual and combined effects of LUCC and CC, at regional and local scales.

Chilean mountain ecosystems are highly vulnerable to climate change fulfilling seven of the nine characteristics defined by the United Nations Framework Convention on Climate Change (UNFCCC) [36,37]. Particularly, the Andean foothills of south-central Chile have a complex topography with vertical climatic heterogeneity. This area has been subject to intense processes of territorial transformation, increasing the interest to study

the vulnerability of their water resources to face forcing factors such as LUCC, CC, and the unsustainable exploitation of forest resources.

The present study was conducted in two watersheds located in the Andean foothill (Quino and Muco watersheds) to determine the effects on the hydrological response under possible future scenarios of LUCC and CC. For this purpose, the SWAT hydrological model was evaluated to assess the impact of these forcing factors on the hydrological response of the Quino and Muco watersheds.

In this way, the present research aimed to (i) model and assess the historical impacts of LUCC on water resources and (ii) assess the individual and combined effects of potential LUCC and CC on water resources. This provides plausible information on the vulnerability of watersheds to the individual and the combined effects of LUCC and climate changes.

## 2. Materials and Methods

### 2.1. Study Area

The study was conducted in watersheds located in the Araucanía region, in the south-central zone of Chile. (38°10′00″ and 38°40′00″ S) (Figure 1). This area is located in a transition zone between Mediterranean and humid temperate conditions, climatic transitional effect eases sclerophyllous species development in its northern section, allowing typical formations of the southern temperate forest [38].These conditions are directly related to the productive use of the territory currently dominated by an extensive area of forest plantations composed mainly offast-growing exotic species. The area is crossed by the Imperial River; it has an industrial sector closely linked to the forestry industry and agricultural production. Within this area, Quino and Muco watersheds were selected (Figure 1), considering the availability of meteorological and fluviometric stations with information available for 35 years (1982–2016).

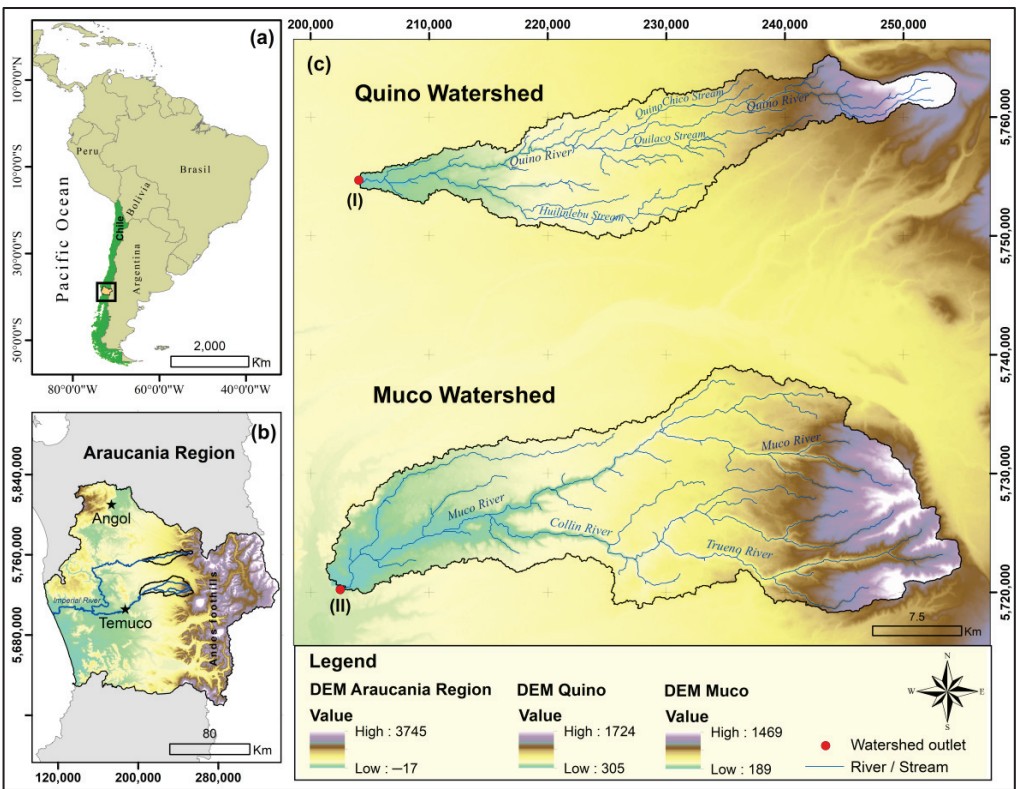

**Figure 1.** Location of study area. Context in South America (**a**), location in Chile (**b**), study watersheds (**c**). Fluviometric Stations: "Rio Quino en Longitudinal" (I) and "Rio Muco en Puente Muco" (II).

The Quino river rises at 1700 m.a.s.l in the Andes Mountains. Its main tributaries from upstream to the watershed outlet are the Quino Chico stream from the northwest, the

Quilaco stream from the north, and the Huilinlebu stream from the southwest. The Quino river watershed covers an area of 299 km$^2$ with an altitude that ranges between 305 and 1724 m above sea level. In addition, the Mucoriver is fed mainly by the Collin River and the Trueno River. Its source is located at 1085 m.a.s.l in the foothills of the Andes, flowing into the Cautín River, near the town of Pillanlelbun. The Muco river watershed has a surface area of 651 km$^2$ with altitudes ranging from 189 to 1469 m.a.s.l. Both basins have a cold and rainy temperate climate with Mediterranean influence. This zone has low temperatures year-round with annual averages ranging between 10 °C and 12 °C for the Quino and Muco basins, as well as a rainfall regime that increases with altitude with annual averages of 1253 mm and 2693 mm, respectively.

### 2.2. SWAT Hydrological Modeling

### 2.2.1. SWAT Model

The Soil and Water Assesment Tool (SWAT), 2012 version, coupled with the ArcSWAT graphic platform was applied to evaluate the effects of LUCC and CC in the study watersheds. SWAT is a semi-distributed model designed to predict the impact of different land uses on the water balance, as well as to evaluate the nutrient and sediment export on watersheds [39]. Several pieces of information were used as input data: on the one hand, the Digital Elevation Model (DEM) was obtained from Alos-1 Palsar images with a spatial resolution of 12.5 m (https://vertex.daac.asf.alaska.edu, accessed on 1 February 2022), on the other hand, the soil type was obtained from the Natural Resources Information Center (CIREN) (https://www.ciren.cl, accessed on 1 January 2020); meanwhile, the climatic information (precipitation and minimum and maximum temperature), was extracted from the Center for Climate and Resilience Research (CR)$^2$ (https://www.cr2.cl, accessed on January 2020) databases with a daily time step (Figure 2). Climatic information was extracted from the CR2MET gridded product with a resolution of 0.05°, which represents a spatially distributed data set developed specifically for Chile [40,41].

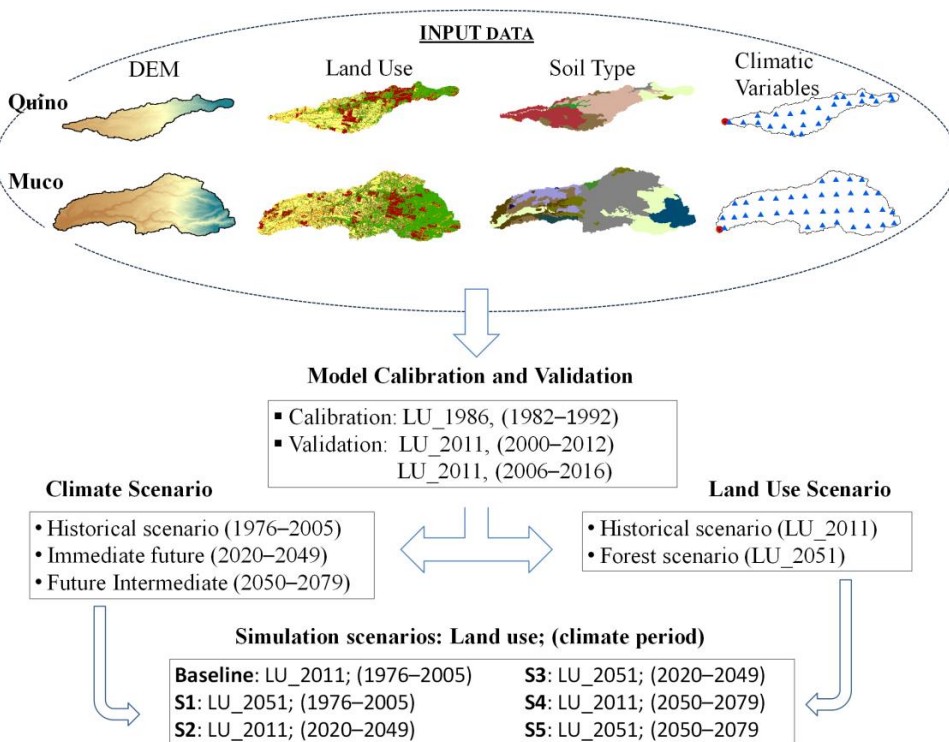

**Figure 2.** Hydrological simulation methodology.

Discretization of watersheds was conducted by splitting them into sub-basins and Hydrological Response Units (HRUs). HRUs are areas with relatively homogeneous land

use and cover conditions; thereby, homogeneous hydrological behavior was also expected. The model response was obtainedonan annual scale. The results of the hydrological process simulation estimate water cycle parameters such as evapotranspiration (ET), percolation (PERC), surface flow (SURQ), lateral flow (LAT_Q), groundwater flow (GW_Q), and water yield (WYLD). Surface runoff estimation was performed in the SWAT model using the runoff curve number (SCS CN). The potential evapotranspiration was determined using the Hargraves method, extensively applied for Chilean watersheds studies, using daily rainfall and daily maximum and minimum temperature as the available input data [7,41–43]. Moreover, runofftracking was modeled using the kinematic wave model and Manning equation to determine the velocity of surface runoff [39].

### 2.2.2. Calibration and Validation

Calibration and validation processes were performed as reported in previous studies [41]. A period from 1982 to 2016 was selected for simulation, considering three years for the model warm-up (1979, 1980, and 1981). For this period, Quino and Muco watersheds were subdivided into 99 and 91 sub-basins and 615 and 983 HRUs, respectively.

On the one hand, during the calibration procedure, for both watersheds, information from 1980 to 1992 was used. On the other hand, during the validation procedure for the Quino watershed, a fluviometric data set from 2000 to 2013 was considered. Meanwhile, observed fluviometric information from 2006 to 2016 was used to validate results for the Muco watershed. Calibration, uncertainty, sensitivity analysis, and also validation processes were performed considering the watershed monthly average flow applying the SUFI-2 algorithm [41,44–46], included in SWAT-CUP software. For this purpose, the most sensitive parameters determined included (i) the groundwater delay, (ii) threshold depth of water in the shallow aquifer required for return flow to occur, (iii) groundwater "revap" coefficient, (iv) average slope length, (v) effective hydraulic conductivity in main channel alluvium, (vi) Manning's "n" value for the main channel, (vii) effective hydraulic conductivity in tributary channel alluvium, (viii) available water capacity of the soil layer, and (ix) the fraction of transmission losses from the main channel that enter deep aquifers as detailed in Martínez-Retureta et al. [41].

In addition, both watershed parametrizations were performed in their respective watershed outlet at the fluviometric stations "Rio Quino en Longitudinal" and "Rio Muco en Puente Muco" (Figure 1c). In this sense, both watersheds presented hydrological fitting evaluated as "satisfactory" and "very good" according to Moriasi et al. [47] classification for efficiency criteria:Nash-Sutcliffe (NS), PBIAS, and determination coefficient ($R^2$) (Table A1).

Once the model was calibrated and validated, different LU and climate scenarios were applied to evaluate their impact on hydrological processes at the watershed scale. Such simulations made it possible to quantify the impact of individual and combined effects of the LUCC and climate change on the components of the hydrological cycle (Figure 2). For this purpose, climate patterns simulated by $(CR)^2$ were implemented [39,48].

### 2.3. Evaluating Historical LUCC

Historical periods of LUCC in the Quino and Muco watersheds were analyzed. LU data for 1986, 2001, and 2011 was obtained from Heilmayr et al. [49]. In this study, a historical trend of LU over the last three decades was performed using Landsat satellite image data. LU categories were generated by applying a supervised classification and maximum likelihood [49]. In the present study, to analyze the LUCC, data were grouped into six categories of LU: native forests, forest plantations, shrublands, agriculture, grasslands, and others. The category named "other" includes LUs with lesser extent in the watersheds. The LU map used as SWAT input for modeling was developed using ArcGIS10.4.1 and the Geographic Information System (GIS). LUCCs were evaluated by examining the relative changes in the extent of the LUs for both watersheds (Quino and Muco) among the study periods (1986, 2001, and 2011).

### 2.4. Effect of Historical LUCC on the Hydrological Response

The impact of LUCC on the hydrological response that occurred during the historical period (1986 to 2011) was evaluated in the Quino and Muco watersheds using annual averages of hydrological parameters such as ET, PERC, SURQ, LAT_Q, GW_Q, and WYLD.

To determine significant differences in the annual hydrological parameters simulated between different LU scenarios (LU_1986 and LU_2011), Student's *t*-test for related samples was realized.

### 2.5. Scenario of Forest Expansion

The future LU scenario was modeled using as baseline three cartographic products developed by Heilmayr et al. [49] corresponding to LUs from Chile's Valparaiso Region to the Los Lagos Region in1986, 2001, and 2011. Several variables such as elevation, slope, soil type, previous land use, roads, water bodies, and towns were analyzed to determine forest plantation expansion within the watersheds. We analyzed the relationships between forest expansion (dependent variable) and forcing factors (independent variable) quantified by a logistic regression model (Equation (1)) [50]. The process was performed using ArcGIS 10.4.1.

$$P(y = 1 \mid x) = \frac{e^{\beta_0 + \sum_{i=1}^{n} \beta_i x_i}}{1 + e^{\beta_0 + \sum_{i=1}^{n} \beta_i x_i}} \tag{1}$$

In order to verify model accuracy, validation was performed after the calibration procedure. LU_2011 scenario was simulated for validation purposes as it was compared to a cartographic product based on satellite images and in/situ observation as developed by Heilmayr et al. [49]. The parameters obtained through the logistic regression model were used to simulate the scenario of future forest expansion until 2051. In particular, the spatial patterns of forest plantation expansion observed within the study watersheds and taking into account the current legislation (law 202831) on native forest recovery were used [51].

The expansion of forest plantations was strongly determined by slope, distance to native forest, presence of native forests, distance to urban centers as well as the presence of agricultural lands, and distance to roads, among other variables (Table 1). Parameters sign (β) of variables related to landscape, indicates that elevated topographies restrict, to a certain extent, forest plantations establishment. However, in the lower elevation sectors, plantations are established in areas with steep slopes. Regarding the distance variables, parameters show that forestry activity takes place near populated centers, roads, agricultural land, and previously established plantations because these areas offer favorable conditions for their establishment presenting suitable infrastructure and equipment facilities that favor forest activity. Likewise, forest plantation expansion is associated with the presence of agricultural activities and native forests. On the other hand, according to the distance to rivers variable, forest expansion seems to occur away from rivers.

**Table 1.** Logistic regression parameters fitted for forest expansion scenario.

| Variable | β(i) | Standard Error | Wald [a] | *p* |
|----------|------|----------------|----------|-----|
| Elevation | −0.0026 | 0.002 | 0.949 | |
| Slope | 0.0844 | 0.0266 | 10.070 | ** |
| Distance to urban centers | −0.0002 | $4 \times 10^{-5}$ | 18.727 | ** |
| Distance to roads | −0.0005 | 0.0002 | 5.206 | ** |
| Distance to native forest | −0.007 | 0.0017 | 16.411 | ** |
| Distance to agricultural lands | −0.0064 | 0.0042 | 2.229 | |
| Distance to forest plantations | −0.0004 | 0.0003 | 1.826 | |
| Distance to rivers | 0.0003 | 0.0002 | 1.413 | |
| Agricultural land presence | 11.5232 | 3.941 | 8.548 | ** |
| Native forest presence | 16.1227 | 3.721 | 18.778 | ** |

Note(s): [a] Wald test was used to evaluate the statistical relevance of every model coefficient (β). (**: $p < 0.05$).

## 2.6. Individual and Combined Effects of LUCC and CC on the Hydrological Response

To estimate the individual and combined influence of LUCC and CC on the hydrological components, an experimental design with two variables was implemented. LU was considered as a variable with two scenarios: LU_2011 and LU_2051 (forest scenario), while the climatic variable considered three scenarios using different meteorological data: from 1976 to 2005 (historical period), from 2020 to 2049 (immediate future), and from 2050 to 2079 (intermediate future). In this way, six combinations were established considering a baseline (LU_2011, historical climatic period) and five possible future scenarios (Table 2).

**Table 2.** Simulated scenarios of land use and climate change.

|  |  | Land Use Scenario | |
|---|---|---|---|
|  |  | **2011** | **2051** |
| Climate escenario | 1976–2005 | Historical Scenario (Baseline) | Scenario 1 (S1) |
|  | 2020–2049 | Scenario 2 (S2) | Scenario 3 (S3) |
|  | 2050–2079 | Scenario 4 (S4) | Scenario 5 (S5) |

Simulated scenarios were compared with the baseline in order to assess the individual and combined impacts of climate change and LU dynamics on water cycle components (ET, PERC, SURQ, LAT_Q, GW_Q, and WYLD). For such purpose Equation (2) was implemented at a watershed scale, thus the absolute variation of the parameters was calculated for each modeled scenario, considering all aforementioned parameters.

$$\Delta_{absolute} = Volume_{Historial\ Scenario} - Volume_{Simulated\ Scenario} \tag{2}$$

In this way, absolute change values of the predicted hydrological parameters could determine the water balance component's sensitivity to likely LU and CC future scenarios. Such results could improve water cycle understanding within the watersheds.

## 3. Results and Discussion

### 3.1. LUCC Scenarios: Historical Period Analysis

LUCC time series analysis from 1986 to 2011 depicted an expansion of forest plantations and agricultural land at the expense of native forest and shrublands occupation decrease (Figure 3, Table A2).This decrease in native forest cover of 23.0% and 13.3% for the Quino and Muco watersheds, respectively, occurred along with shrubland areas decrease by 3.5% and 7.4%. Shrublandswere also replaced by fast-growth forest plantations covering 21.3% of the Quino watershed and 14.2% of the Muco watershed by 2011. This was an important change considering that such territories possessed forest plantation coverage of less than 4.0% at Quino and 2.0% at Muco watersheds in 1986. In this sense, forest plantations increased from 10.4 km$^2$ in 1986 to 63.6 km$^2$ in the Quino watershed and from 10.2 km$^2$ to 92.8 km$^2$ in the Muco watershed.

The area of forest plantation increased for both watersheds at middle altitudes from upstream to downstream (Figure 3). Such a pattern was related to the proximity to urban areas, roads, and highways along with the location of pulp and construction industries. In addition, the higher altitudes of the watersheds were still mainly covered mainly by native forests and shrublands in 2011 (Figure 3).

Agricultural land also increased in both watersheds, mainly at lower altitudes (Figure 3). Specifically, agricultural land increased by 4.6% from 1986 to 2001 in the Quino watershed, and in the full period (1986–2011), it almost doubled the area, increasing by 8.5%. In the Muco watershed, during the first period (1986–2001), the agricultural land increased by 7.4% but remained the same from 2001 to 2011 (Figure 3, Table A2). In addition, native forests and forest plantations showed the largest relative changes over the whole period.

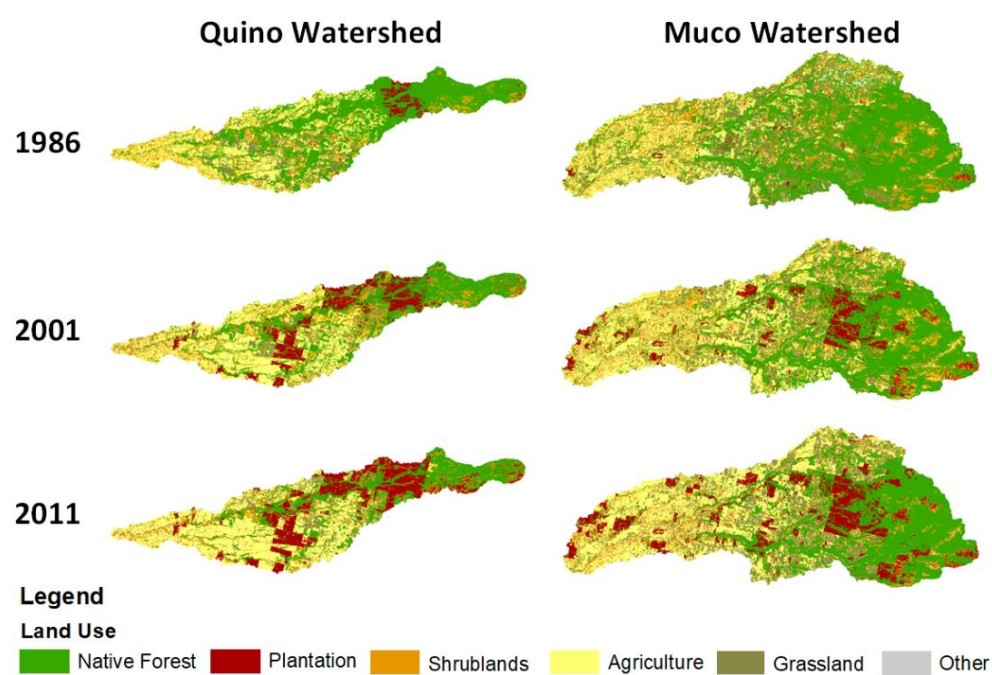

**Figure 3.** Spatial distribution of land use and covers (1986, 2001, and 2011).

*3.2. LUCC Impact on the Hydrological Response: Historical Period Analysis*

Yearly averages of the hydrological parameters for the period from 1982 to 2016 parameters obtained by SWAT simulation are analyzed in Figure 4 considering LU_2001 as a transition period for this study. ET presented significant increasing trends from 1986 to 2011 land uses, with absolute changes in the annual average of 15.3 mm and 25.9 mm for Quino and Muco watersheds meaning a 2.4% and 4.2% relative change, respectively (Figure 4a,b, Table A2). An increase in ET could be directly related to forest plantation expansion considering that this LU covered 21.3% and 14.2% by 2011 with percent increments of 17.8% and 12.7% from 1986 for the Quino and Muco watersheds, respectively. This behavior was also observed by Shi et al. [3] who suggest that effects ofLUCC could take place mainly in ET, the soil water retaining capacity, and water interception capacity by plants.

Statistically significant increasing trends were also reported for both watersheds in SURQ and LAT_Q parameters. In the Quino watershed, mean yearly values reached 45.9 mm and 10.1 mm for such parameters, respectively, representing 14.1% and 11% of relative change among the analyzed LUs (Figure 4a). On the other hand, the Muco watershed presented lower relative changes (0.2% and 6.3%) for SURQ and LAT_Q parameters, respectively (Figure 4b).

Conversely, PERC presented a significant decrease with LU_2011 if compared to LU_1986 for both study watersheds. Average annual changes of 76.8 mm for the Quino watershed and 83.4 mm for Muco occurred; representing 11% and 12.2% relative change, respectively (Figure 4a,b). Thereby, decreasing trends in soil water infiltration also caused lower GW_Q values at Quino and Muco watersheds. Significant differences were obtained for the GW_Q parameter for both watersheds, with absolute change values reaching 99.6 mm and 59.7 mm for 15% and 10.1% of relative change, respectively.

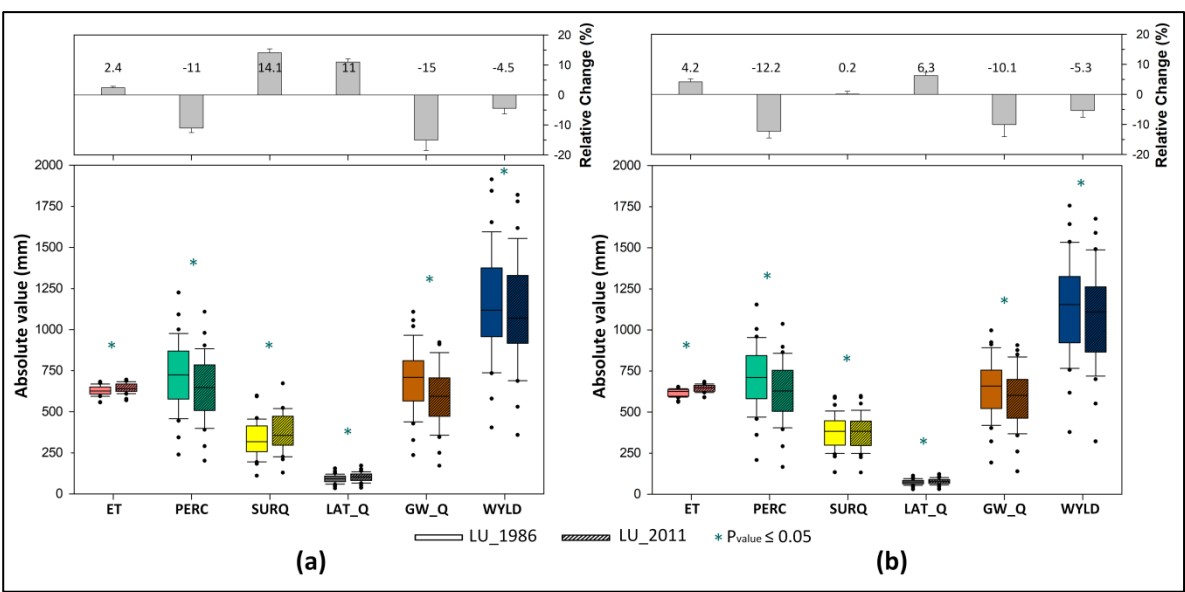

**Figure 4.** Annual values of hydrological cycle parameters from (**a**) Quino and (**b**) Muco watershed: LU_1986 and LU_2011 scenarios (lower box). Relative changes between LU (upper box).

Results reveal that an increase in surface runoff and a decline in groundwater recharge could be associated with the spread of forest plantation (17.8% and 12.7%) and agricultural lands (8.5% and 7.4%) as well as with the loss of native forests (23% and 13.2%) and, to a lesser extent with shrublands (3.5% and 7.4%) in the study watersheds. In this case, it could be considered that the massive conversion of LUs from native forests, to both fast-growth exotic plantations and agriculture, occurred in the study area could lessen soil infiltration capacity, thereby causing a high percentage of rainwater to turn into surface runoff [28,52] and, another percentage to be absorbed directly by plantations, leading to observed decrease of GW_Q and WYLD (Figure 4a,b) [28,53]. Similar evidence but toa higher extent wasrecently found in the Fichaa watershed in Ethiopia where, due to extensive LUCC, surface water sources dried up and water levels at wells decreased significantly [17].

Finally, water yield at both watersheds kept a decreasing significant trend with yearly relative averages changes ranging from 4.5% and 5.3% during the period. Such reductions represent, in absolute values, a lowering of 47.2 mm and 55.4 mm on the yearly average water yield at Quino and Muco watersheds, respectively (Figure 4a,b).

### 3.3. Hydrological Response to LUCC and CC: Individual and Combined Effects

#### 3.3.1. Forest Expansion Scenario

According to the logistic regression method performed, forest plantation expansion is projected mainly in the middle and lower altitudes in the study watersheds, including also some areas at higher elevations, covered mainly by shrublands (Figure 3). In addition, forest expansion innative-forest-covered zones was not considered for a future scenario. This conversion has taken place not only in the study watersheds but in almost all the south-center zone of Chile [28,38,49,54], however, current forest legislation generated a regulatory framework encouraging native forest protection through management and conservation actions [51]. In this way, it is supposed that the areas of native forest present in 2011 will not be altered in the future under the current legislation (Figure 5).

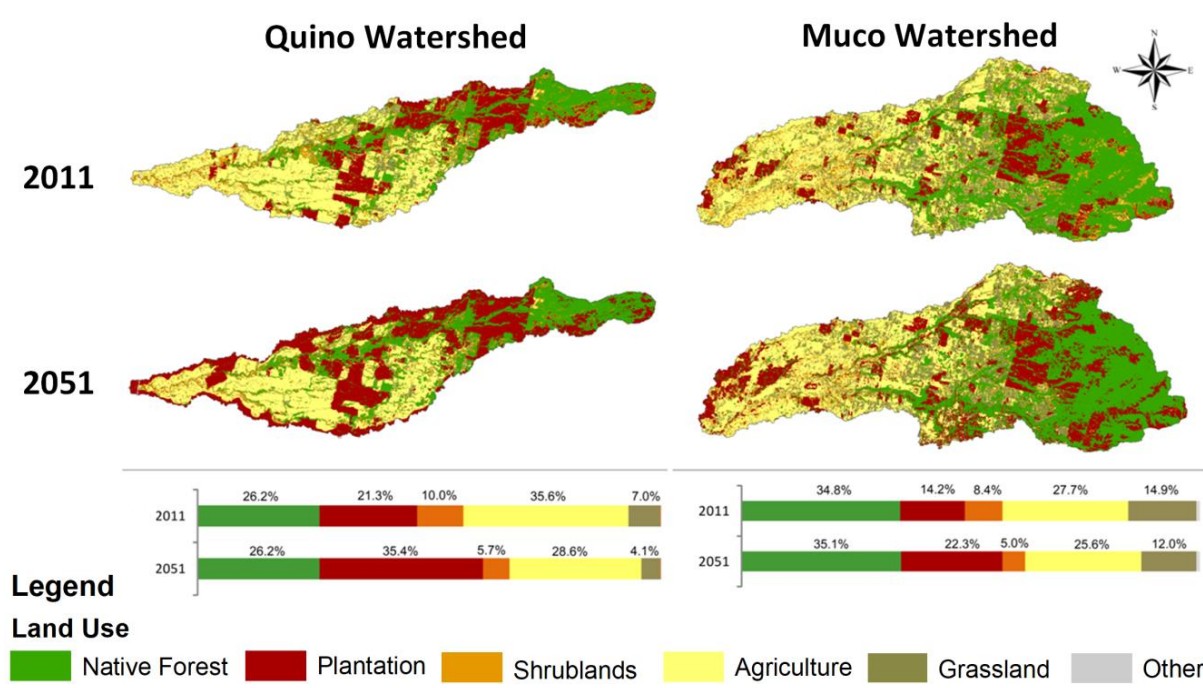

**Figure 5.** Spatial representation (**upper**) and percentage LUCC area (**lower**) for LU_2011y LU_2051 at theQuino and Muco watersheds.

Projected changes in the Quino and Muco watersheds showed a significant increase in forest plantations by 14.1% and 8.1% to cover 22.3% and 35.4% of their areas in 2051, respectively. New forest plantation areas consider the conversion of agricultural lands (7.0% and 2.1%), shrublands (4.3% and 3.4%), and grasslands (2.9% and 2.9%) in the Quino and Muco watersheds, respectively.

### 3.3.2. Future Climate Change Projections

Future climate projections depicted in our previous research [41] were used to quantify the effect of CC on water resources in the study watersheds. It is expected that the temperature would increase by 0.8 °C in the immediate future scenario (2020–2049) at both watersheds; meanwhile, the temperature increase would reach 1.5 °C in the intermediate future (2050–2079). Conversely, precipitation projections depicted lowering trends with yearly average decreases ranging from 37 mm and 127.0 mm for Quino and Muco watersheds, respectively, in the immediate future scenario and 42.0 mm and 140.0 mm in the intermediate scenario.

### 3.3.3. Hydrological Response Facing Future Scenarios

Similar results for both study watersheds were obtained under individual and combined effects of LUCC and CC (Figures 6 and 7, Table A3). Such response could be attributed to similar climatic and physical-geographical characteristics at both watersheds.

According to modeled scenarios, an increasing trend in ET could be expected; however, for this parameter, individual CC effects (S2 and S4) seemed to present higher consequences if compared to the LUCC effect (S1) (Figures 6a and 7a). Nevertheless, as presented by S3 and S5 results, ET would be enhanced by combined LUCC and CC effects presenting relative changes of this parameter ranging from 4.75% and 8.26% at Quino watershed and 4.10% and 8.26% at Muco watershed, for the immediate and intermediate future (Figures 6a and 7a, Table A3). In addition, percolation (PERC) presented a considerable increase under S1, meanwhile, the opposite effect was observed for both CC scenarios S2 and S4 (Figures 6b and 7b). A lowering trend was observed for this parameter with a higher impact on the intermediate future with relative changes ranging between−5.27%

and −5.81% for the Quino and Muco watersheds, respectively (Table A3). However, the combined effect of both forcing factors seemed to be overlapped as LUCC softens CC effects over percolation at Quino and Muco.

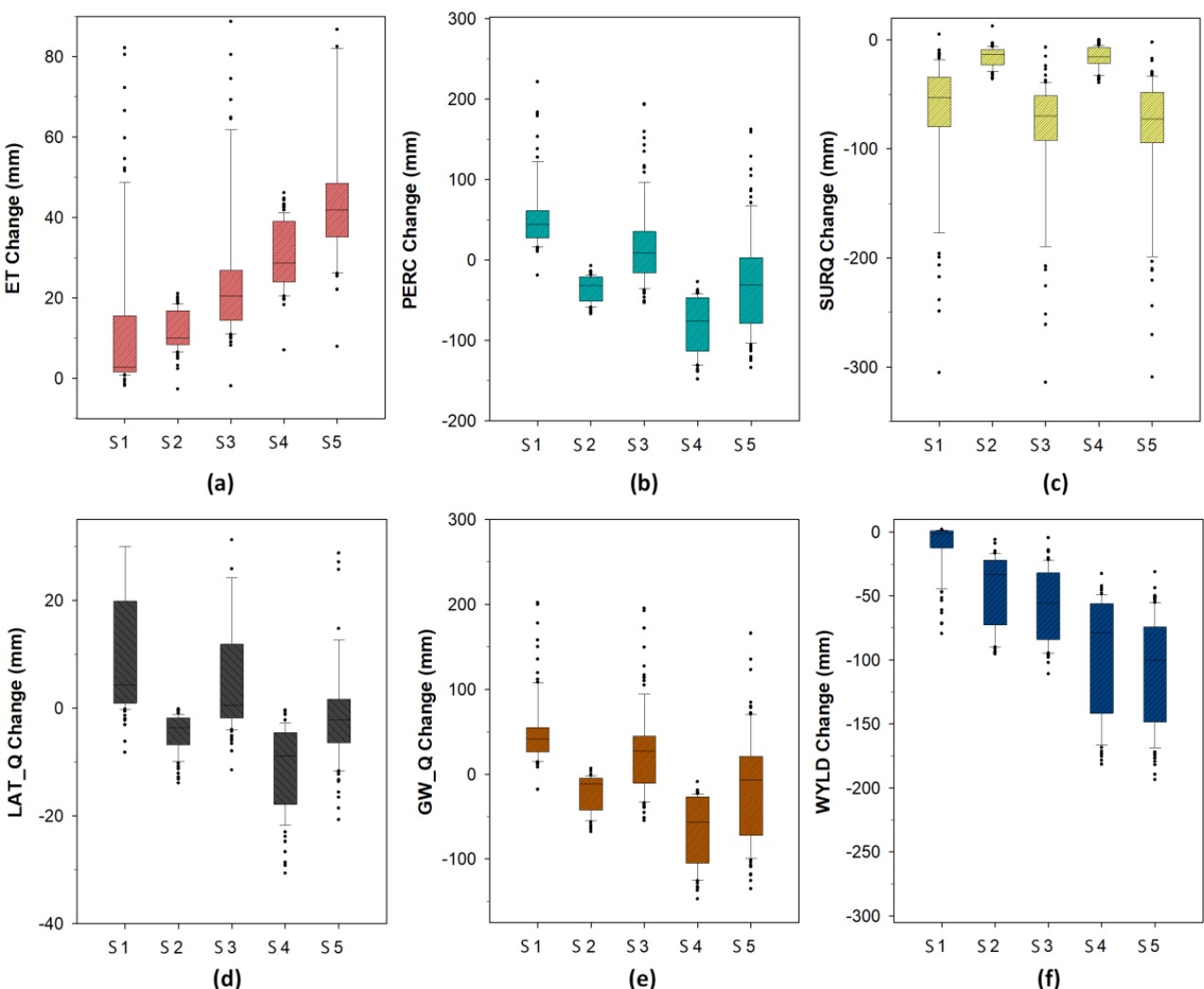

**Figure 6.** Individual and combined effects of land use/cover change and climate change at Quino watershed on the ET (**a**), PERC (**b**), SURQ (**c**), LAT_Q (**d**), GW_Q (**e**) and WYLD (**f**). (Horizontal scale: S1: LU2051_Historical; S2: LU2011_Immediate Future; S3: LU2051_Immediate Future; S4: LU2011_Intermediate Future; S5: LU2051_Intermediate Future).

From the surface runoff point of view (Figures 6c and 7c), both forcing factors seemed to cause decreasing trends at both study watersheds. However, the land use and cover change scenario presented the largest individual effect, implying percent changes ranging between −10.37% and −15.57% in theQuino and Muco watersheds, respectively (Table A3). This result could be related to the forest plantation increase that considered 35.4% and 22.3% of the total area at Quino and Muco watersheds, respectively (Figure 5). On their combined effects, CC enhances LUCC effects (S3 and S4) as projected climatic scenarios foresee an important temperature increase with fewer precipitations for both watershed studies.

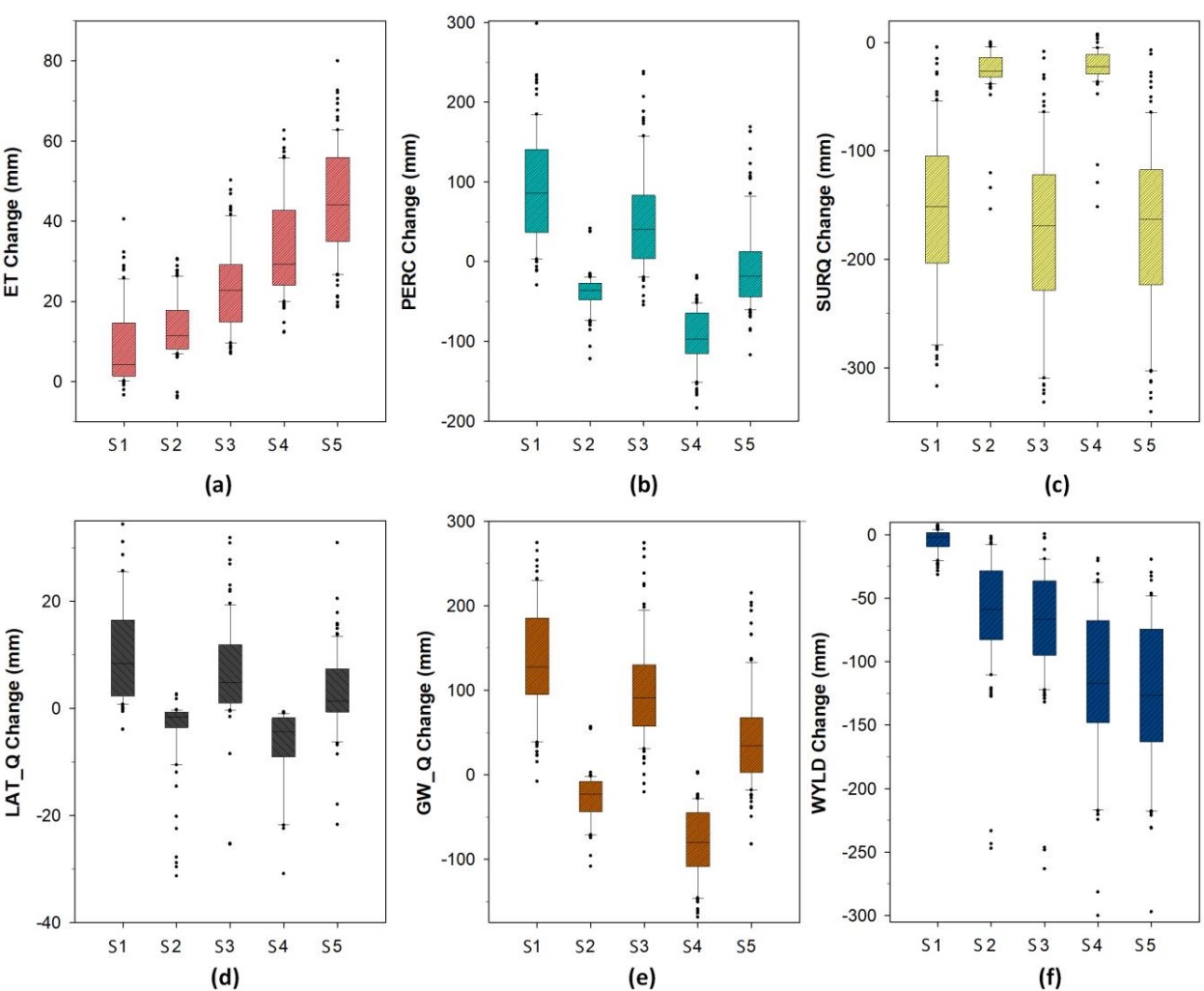

**Figure 7.** Individual and combined effects of land use/cover change and climate change at Muco watershed on the ET (**a**), PERC (**b**), SURQ (**c**), LAT_Q (**d**), GW_Q (**e**) and WYLD (**f**). (Horizontal scale: S1: LU2051_Historical; S2: LU2011_ Immediate Future; S3: LU2051_ Immediate Future; S4: LU2011_ Intermediate Future; S5: LU2051_ Intermediate Future).

Similar trends were observed for the remaining hydrological cycle variables analyzed (LAT_Q and GW_Q) for all the scenarios modeled at both watersheds. In the Quino watershed, LAT_Q and GW_Q showed a favorable effect with changes of 8.51% and 9.61% under scenario S1, while in the Muco watershed the parameters reached 13.55% and 14.90%, respectively (Table A3). CC scenarios (S2 and S4) caused an opposite impact in both study watersheds. The combined effect between the LUCC and the CC scenarios (S3 and S5) shows a favorable compensation for both parameters responding to a bigger influence of the LUCC can be observed. These results were related to an increase in forest plantations and a decline in agricultural and shrubland areas, projected for the LUCC future scenario (LU_2051).

Finally, the water resource yield (WYLD) presented a marked decreasing trend caused by both stressors. In their combined effect (S3 and S5), the water yield decrease was increased even further, presenting relative changes of −3.18% and −6.18% for the Quino watershed and −2.80% and −5.53% for the Muco watershed (Table A3).

The high sensitivity of water balance components to CC has been reported in other studies worldwide [3,30,55–57]. Shi et al. [3], in a study conducted at HuaiRiver upstream, in China, reported that LUCC and CC have jointly affected water resources by increasing surface flow and evapotranspiration. In this sense, CC effects on the hydrolog-

ical response were compensated by LUCC effects [3]. However, different impacts were observed [20,57–59] by LUCC and climatic variability on the hydrological process. In addition, a combined study of LUCC and CC in the Hoeyariver watershed of Korea by Kim et al. [57] also showedthat combined effects were similar to CC scenarios, suggesting this forcing factor presents a higher influence.

Another study developed at the Finchaa watershed in Ethiopia [35] revealed that, under LUCC and climatic changes, the soil moisture required for crop growth becomes reduced, and then land degradation problems occurred with severe landslides. In addition, Vlek et al. [59] further reported that land degradation is related also to deteriorating climatic conditions and human intervention.

We found that ET was sensitive to CC, and its effect was exacerbated under a LUCC scenario. Meanwhile, the SURQ presented a greater sensitivity to the LUCC effect, worsening its unfavorable projection when incorporating climate change scenarios. On the other hand, PERC, LAT_Q, and GW_Q showed a higher sensitivity to the LUCC, which is compensated by the effect of CC.

Finally, the variability of SURQ, LAT_Q, and GW_Q parameters in the watersheds induces that WYLD is also affected by both forcing factors. Under CC scenarios a lessening in these three parameters is projected; meanwhile, the LUCC scenario only negatively affects the SURQ parameter. This leads to the fact that the WYLD presents a high sensitivity to CC that can be aggravated by the LUCC. This should be considered before taking relevant decisions when evaluating the implementation of measures to mitigate CC effects in the region.

## 4. Conclusions

This study allowed us to distinguish the individual and combined effects of LUCC and CC on the components of the water balance in the Quino and Muco basins in the south-central zone of Chile. From the LUCC point of view, the main changes are manifested in a slight increase in ET and a considerable decrease in SURQ; however, the LUCC affects to a lesser extent the WYLD by favoring the LAT_Q and GW_Q. On the other hand, CC increases ET to a greater extent and leads to considerable decreases in LAT_Q and GW_Q and lesser effects on the rest of the hydrological parameters, leading to a greater impact than LUCC on the WYLD. When both global changes act synergistically, the decrease in the WYLD of the watersheds becomes worse.

The results of the present investigation, developed within two hydrographic watersheds located in the south-central zone of Chile, could be used as evidence for mitigating the negative effects of climate change on water resources. In this way, new strategies for the preservation and rehabilitation of native forests should be implemented, since this has a relevant role in the water dynamics within the watersheds, allowing the reduction of the water velocity, increasing the water infiltration and storage in the soil and then contributing to rivers and other water bodies. Such strategies of watershed management could then contribute tothe recovery of the watersheds structure and thus its function, also recovering the ecosystem services of regulation, storage, and water supply. This could finally contribute to decision-making on future watershed management leading to the fulfillment of sustainable development goals.

The hydrological model shows an adequate evaluation of the combined impact of the LUCC and CC, allowing its reproduction in other areas of interest with similar physical-geographical conditions. However, the availability and quality of hydroclimatic data in the region must be improved to facilitate the understanding of current global changes and predict future changes. In this way, we highlight the need to advance in regional development and cooperation to promote management strategies resilient to LUCC in order to counteract the imminent effects of CC in the watersheds.

**Author Contributions:** Conceptualization, R.M.-R. and M.A.; methodology, R.M.-R., M.A. and N.J.A.; software, R.M.-R.; validation, R.M.-R.; formal analysis, R.M.-R., M.A. and N.J.A.; investigation, R.M.-R., M.A., N.J.A., O.L. and R.O.B.; resources, R.M.-R., M.A. and N.J.A.; data curation, R.M.-R., L.R.-L. and I.D.-L.; writing—original draft preparation, R.M.-R., M.A. and N.J.A.; writing—review and editing, R.M.-R., M.A., N.J.A., R.U., C.E., O.L., L.R.-L., I.D.-L. and R.O.B.; visualization, R.M.-R., N.J.A. and I.D.-L.; supervision, M.A., N.J.A., R.U., C.E., O.L. and R.O.B.; project administration, R.M.-R., M.A., O.L. and R.O.B.; funding acquisition, R.M.-R., N.J.A. and M.A. All authors have read and agreed to the published version of the manuscript.

**Funding:** This research was funded by CRHIAM Project (ANID/FONDAP/15130015), ANID/National Doctorate Scholarship 2016 Grant N° 21160323 and Project ANID/Postdoctoral FONDECYT/3220382.

**Institutional Review Board Statement:** Not applicable.

**Data Availability Statement:** Not applicable.

**Acknowledgments:** Rebeca Martínez-Retureta and the authors are grateful to the National Agency for Research and Development (ANID) of the Chilean Government for the funding provided by the National Doctorate Scholarship 2016 Grant N° 21160323, the CRHIAM Project (ANID/FONDAP/15130015) and Project ANID/Postdoctoral FONDECYT/3220382. Further, for the support provided by the University of Concepcion and ECOLAB laboratory UMR 5262. Authors especially thank the Center for Climate and Resilience Research (CR)[2] for providing the meteorological data.

**Conflicts of Interest:** The authors declare no conflict of interest.

## Appendix A

**Table A1.** Calibration and validation of total monthly flow values and their classification.

| | Quino Watershed | | | |
|---|---|---|---|---|
| | **Calibration** | | **Validation** | |
| $R^2$ | 0.88 | Very Good | 0.89 | Very Good |
| NSE | 0.84 | Very Good | 0.79 | Very Good |
| PBIAS | −3.11% | Very Good | −20.68% | Satisfactory |
| | **Muco Watershed** | | | |
| | **Calibration** | | **Validation** | |
| $R^2$ | 0.88 | Very Good | 0.92 | Very Good |
| NSE | 0.88 | Very Good | 0.89 | Very Good |
| PBIAS | 5.92% | Very Good | −11.85% | Good |

**Table A2.** Changes in land use and cover types for the observed study years.

| Land Use | Land Use and Cover (%) | | | Changes in Land Use and Cover (%) | | |
|---|---|---|---|---|---|---|
| | **1986** | **2001** | **2011** | **1986–2001** | **2001–2011** | **1986–2011** |
| | Quino Watershed | | | | | |
| Native forest | 49.1 | 33.4 | 26.2 | −15.8 | −7.2 | −23.0 |
| Plantation | 3.5 | 13.9 | 21.3 | 10.5 | 7.3 | 17.8 |
| Shrubland | 13.5 | 14.8 | 10.0 | 1.3 | −4.9 | −3.5 |
| Agriculture | 27.1 | 31.6 | 35.6 | 4.6 | 3.9 | 8.5 |
| Grassland | 6.8 | 6.0 | 7.0 | −0.7 | 0.9 | 0.2 |
| Other * | 0.0 | 0.2 | 0.1 | 0.1 | −0.1 | 0.0 |
| | Muco Watershed | | | | | |
| Native forest | 48.1 | 38.4 | 34.8 | −9.7 | −3.6 | −13.3 |
| Plantation | 1.6 | 9.1 | 14.2 | 7.5 | 5.2 | 12.7 |
| Shrubland | 15.8 | 13.5 | 8.4 | −2.3 | −5.1 | −7.4 |
| Agriculture | 20.3 | 27.7 | 27.7 | 7.4 | 0.0 | 7.4 |
| Grassland | 13.6 | 10.8 | 14.9 | −2.8 | 4.1 | 1.3 |
| Other * | 0.7 | 0.6 | 0.0 | −0.1 | −0.6 | −0.7 |

Note(s): * Category "other" considers water bodies, urban territories, stubby, and wetlands.

**Table A3.** Relative change of modeled water cycle parameters for Quino and Muco watersheds.

| | Annual Average Relative Change (%) | | | | |
|---|---|---|---|---|---|
| | **Quino Watershed** | | | | |
| | **Scenario 1** | **Scenario 2** | **Scenario 3** | **Scenario 4** | **Scenario 5** |
| ET | 2.03 | 2.40 | 4.75 | 5.27 | 8.26 |
| PERC | 9.69 | −4.11 | 5.25 | −9.52 | −0.42 |
| SURQ | −10.37 | −2.73 | −13.05 | −2.38 | −12.64 |
| LAT_Q | 8.51 | −3.45 | 4.82 | −7.92 | 0.07 |
| GW_Q | 9.61 | −2.05 | 7.33 | −7.45 | 1.62 |
| WYLD | −0.70 | −2.37 | −3.18 | −5.26 | −6.18 |
| | **Muco Watershed** | | | | |
| | **Scenario 1** | **Scenario 2** | **Scenario 3** | **Scenario 4** | **Scenario 5** |
| ET | 1.50 | 2.35 | 4.10 | 5.81 | 7.78 |
| PERC | 8.66 | −3.87 | 4.50 | −9.74 | −1.64 |
| SURQ | −15.57 | −2.12 | −17.37 | −1.76 | −16.91 |
| LAT_Q | 13.55 | −3.14 | 10.02 | −7.93 | 4.81 |
| GW_Q | 14.90 | −2.40 | 11.83 | −8.15 | 5.80 |
| WYLD | −0.30 | −2.35 | −2.80 | −4.98 | −5.53 |

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
