# Peer review of "Influence of Climate and Land Cover/Use Change on Water Balance: An Approach to Individual and Combined Effects"

_water, doi:10.3390/w14152304_

Round 1

Reviewer 1 Report

The paper estimates impacts of climate change, land use and land cover changes on altering elements of water balance with the use of selected hydrological models. Two watersheds located in the south-central part of Chile were selected as the case studies. In my opinion, the study is interesting and significant both from the theoretical and practical point of view. Thus, I would recommend the paper for publication in the journal. While I don’t have any particular comment related to the methodology, I would recommend to make the following amendments in the paper:

1. Please provide more detailed description of the climatic and hydrological conditions of the study area.

2. Please clarify the time step (daily, monthly, other?) of the meteorological and hydrological data used in your study.

3. Please improve the linguistic quality of the paper. Please avoid using the Spanish words in the text, as for example on p. 3, line 115.

With regard to the aforementioned remarks, it is recommend to accept the paper for publication after minor corrections.

Author Response

Dear reviewer.

We appreciate your comments as it helped us to improve the quality of the manuscript´s discussions as well as its understanding. Thank you.

Please, find in the attached document the answers.

Best regards.

Reviewer 2 Report

The suggestions for improvement is attached 

Author Response

(The authors gave the same response as above.)

Reviewer 3 Report

In the manuscript entitled “Influence of climate and land cover/use change on water balance: an approach to individual and combined effects” the authors attempt to predict future cover/use changes considering a forest expansion scenario according to Chilean regulations. In this way an expansion by 42.3km2 and 52.7km2 at Quino and Muco watersheds respectively was predicted, reaching a watersheds’ occupation of 35.4% and 22.3% at 2051. Additionally, local climatic model RegCM4-MPI-ESM-MR was used considering periods from 2020-2049 and 2050-2079, under RCP 8.5 scenario. Finally, SWAT model was applied to assess the hydrological response of both watersheds facing the considered forcing factors. Five scenarios were determined to evaluate LUCC and CC individual and combined effects. The results are valuable and have important policy implications. I have only few comments and suggestions for the authors:

Please supplement the literature on global phenomenon of LUCC. See below for example:

Rajpar, H., et al. (2019). "Agricultural Land Abandonment and Farmers’ Perceptions of Land Use Change in the Indus Plains of Pakistan: A Case Study of Sindh Province." Sustainability 11(17): 4663.

Please explain what the dependent variable in the logistic regression model is.

The conclusions section needs to be enhanced while providing the policy recommendations emerging from the results of the study.

Please also elaborate on the limitations of your study and provide directions for future research.  

Author Response

(The authors gave the same response as above.)
